# Wave Climate and the Effect of Induced Currents over the Barrier Reef of the Cays of Alburquerque Island, Colombia

**Serguei Lonin *** , **Carlos Alberto Andrade and Julio Monroy**

Grupo de Investigación en Oceanología, Escuela Naval de Cadetes "Almirante Padilla", Isla Naval Manzanillo, Cartagena 130001, Colombia; candrade@exocol.com (C.A.A.); julio.monroy@armada.mil.co (J.M.)
* Correspondence: slonin@costa.net.co

**Abstract:** The Alburquerque Cay Islands belong to a group of western Caribbean atolls, with a barrier reef of coral that is more than 8 km in diameter. An understanding of the reef ocean dynamics and its direct relationship to the functioning of the ecosystem is important since the significant energies that are involved should play a fundamental role in the coral and fish distributions in the coral reef, which are fundamental for its sustainability. The microtidal regime and the predominance of the trade winds produces coastal circulation, which is induced by waves breaking against the barrier. The study of the water dynamics that are described in this paper was carried out by using a stationary model that is based on the shallow-water equations theory, with consideration to the radiation stress gradients in the waves, whose patterns were evaluated with the pseudo-data of a reanalysis of a virtual buoy in front of the atoll and were propagated in the domain of interest by using the simulating waves nearshore (SWAN) spectral model. The results demonstrate the bi-modal behavior of the currents of water jets, with the occurrence of extreme waves with velocities greater than 1 m/s along the barrier. For the mean wave regime, circulation occurred around the coral mounds. This circulation suggests that the reef, the orientation of which coincides with the direction of the predominant waves, was formed under these average conditions. The wave climate is primordial to the reef environment, and climate change may affect the coral health.

**Keywords:** Caribbean Sea; Alburquerque Cay Islands; wave-induced currents; SWAN spectral model; coral reef; wave radiation stress gradients

## 1. Introduction

Numerical ocean models have become increasingly valuable tools as we strive to understand the nature of an ocean's dynamics. These models have been developed from more crude and idealized tools to capture much of the complexity of the real ocean [1].

The application of numerical techniques for solving coastal problems by using models is a reliable, cost-effective, and time-saving tool (e.g., [2]). Thus, the proper determination of the wave-parameter selection of the grid structures, advanced methods for conservation in highly nonlinear systems, inverse methods, and other ideas for modern ocean modeling stimulate interest. In this article, a different numerical model was applied to a complex solid reef wall at the Alburquerque Bank by using several steps to improve the accuracy of the numerical models (i.e., a model that had not been previously applied in studies of reef walls).

The Alburquerque Bank is an atoll of a total area of 63.8 km$^2$ that is located between 12°08′–12°12′ N and 81°49′–81°54′ W, which is about 35 km SW of San Andrés Island in the San Andrés, Providence, and Santa Catalina Archipelago (SPSA), Colombia. This bank has a circular shape that encompasses a pre-reef terrace. The east–west diameter exceeds 8 km. The atoll has two cays (Cayo del Norte and Cayo del Sur or Pescadores), with a combined emerged area of 0.1 km$^2$. Alburquerque is permanently occupied by military personnel, and it receives continuous visits from fishermen performing their fishing tasks.

Figure 1 shows the geographical location of the island. Figure 2 is an aerial photograph of the island that shows the two emerged cays, the break zone, the waste sand bands from the break, and the internal lagoon. Because of its ecological relevance, the SPSA, including Alburquerque, was declared a biosphere reserve by the United Nations Educational, Scientific and Cultural Organization (UNESCO) in the year 2000. This reserve is named "Seaflower" and it is the most extensive biosphere reserve in the world, with a surface area of 180.000 km$^2$ [3]. This territory is also immersed in a legal dispute in which Nicaragua instituted proceedings against Colombia in 2001 before the International Court of Justice, which confirmed Colombia's sovereignty over all of the islands in the archipelago, but drew maritime boundaries in favor of Nicaragua [4].

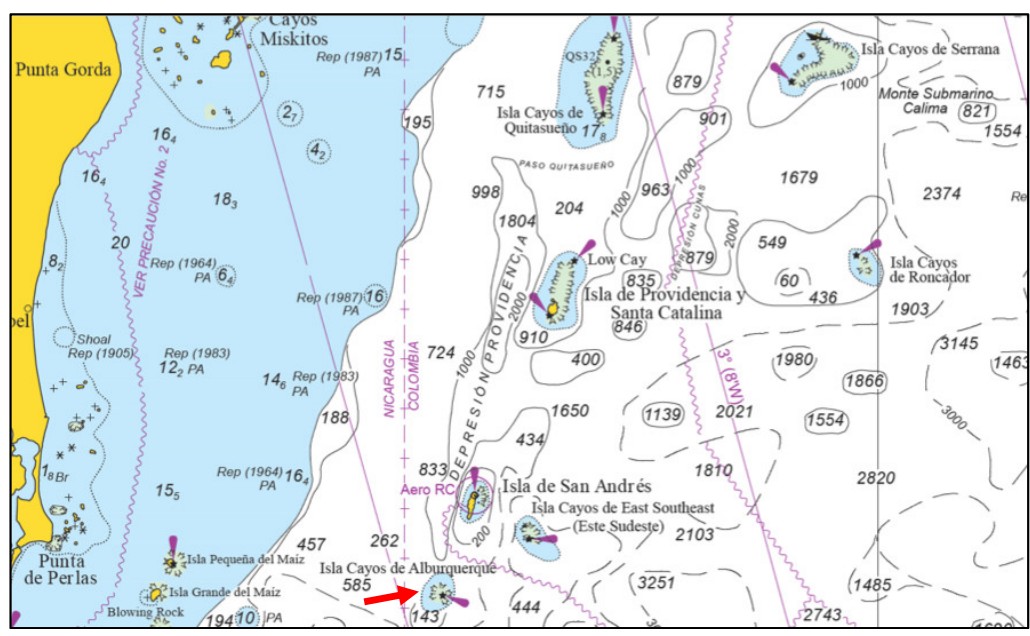

(a)

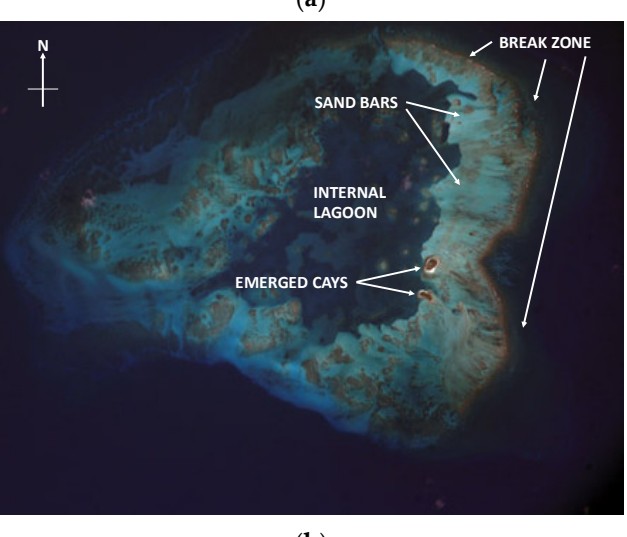

(b)

**Figure 1.** *Cont.*

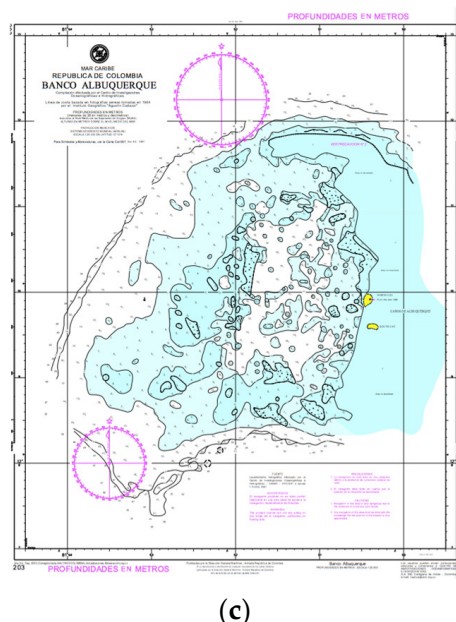

**(c)**

**Figure 1.** (**a**) The spatial location of the cays of Alburquerque Island alongside the general context on the nautical chart COL 007. (**b**) Satellite image (taken from www.oceandots.com, accessed on 15 March 2021) and (**c**) the bathymetry in the nautical chart COL 203 for the Bank of Alburquerque in the Colombian Caribbean.

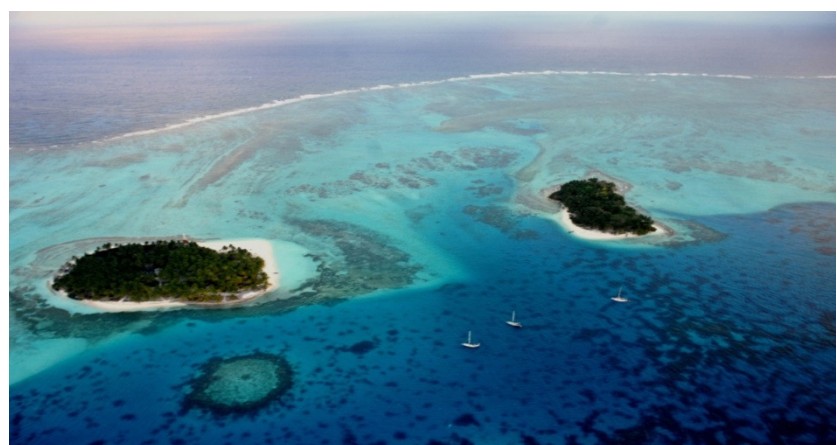

**Figure 2.** Aerial photograph of the cays of Alburquerque Island.

The history of the island dates back to the end of the Triassic Period, which is about 80 million years ago, and it is related to the origin and formation of the Central American and Caribbean Sea elevation [5]. The regional tectonic scheme of the seabed is characterized by fracture zones, the most conspicuous being the San Andrés bill. All the islands, atolls, and coral banks of the archipelago originated from volcanoes that are arranged along the tectonic fractures of the submarine crust, and that are predominantly oriented toward the NNE and SW.

The intertropical convergence zone (ITCZ) dominates the climate over the San Andrés Islands, which modulates the direction and the intensity of the trade winds. These winds are characterized by great uniformity in their velocities. Temporal oscillations over weeks with the passage of cold fronts from the west, or with the passing of atmospheric waves from the east, produce seasonal oscillations. During the windy season, winds from the northeast (NE) and the east (E) predominate, with the infrequent occurrence of storm winds. The climate in Cayo Alburquerque is relatively dry and it follows the typical seasonality of the Western Caribbean, with a windy season from December to March, a transitional

season until June, the "summer of San Juan" in July–August, and a rainy season from September to November [6]. The trade winds are the prevailing direction of the winds in the archipelago area. The trade winds blow from the east–northeast (ENE) direction, with monthly average speed variations between 4m/s (May, September–October) and 7 m/s (December–January, July) [7,8], and with this being the regime that modulates the incident waves that we want to study, taking into account that Cayo Alburquerque is also in the path of hurricanes when their trajectories enter the Caribbean Sea [9,10].

In the dry and windy season, the ITCZ is located further south (0–5° S). Therefore, north winds dominate in this period, with an average speed of 8 m/s, and a daily maximum of 15 m/s. During the Veranillo, in July, winds reach speeds of up to 12 m/s. An increase in calm conditions at sea occurs during this transition period, before the start of the rainy season, as northeast winds inhibit precipitation.

During the wet season, from August to October, the intensity of the medium winds decreases substantially, but very strong showers, which are called "culoe'pollos", are frequent. The decrease in the wind intensity causes the temperature to rise throughout the region.

The predominance of the trade winds throughout this region suggests, on the one hand, that coastal circulation, which is directly related to the amount of space surrounding the barrier reef, is the result of the shape of this barrier and its orientation. On the other hand, the wave regime plays an important role in the formation of this barrier. It seems, this interrelation would be impossible without considering wave-induced currents as one of the main mechanisms of the local circulation.

The interest in examining the variability in the waves of the Alburquerque Cay Islands is due to the fact that, since between 1961 and 1993, an increase of $1.8 \pm 0.5$ mm/year in the mean sea-level rise was registered, and, in the decade from 1993 to 2003, there was an increase of $1.3 \pm 0.7$ mm/year, which was caused by the global warming that was experienced by the planet [11]. This report adds that, although this increase was not uniform, the trend is to increase in the long term. Several studies that were carried out in the Caribbean Sea have produced results with higher values, with major and potential effects on the coastal regions of the archipelago cays [10]. This rise in the sea level means that the effect of the waves is increasingly important in the emerged parts of the low-lying islands, such as Alburquerque.

Because of their relatively low heights, it is known that the cays experience considerable changes in their morphologies, which are due to the retreat of their coastlines, as is found in Cayo Serranilla Island [12] and Cayo Serrana Island [13], which is due to the mean sea-level rise.

The Third National Communication determined that the Colombian ecosystems that are the most vulnerable to the effects of climate change would be those of high mountains, the coastline, and the Caribbean islands [14], which is due to the low altitude of the cays of the archipelago of San Andrés, and to the combined action of the rise in the sea level and the increase in the cyclonic activity that is projected for the Colombian Caribbean. In general, the Cays, Banks and the islands of the archipelago are vulnerable to suffering the retreat of their coastlines, and even submergence above sea level, if the appropriate mitigation measures are not taken to deal with this problem, which is, in part, the object of the study.

There are reports of tropical storm tracks that influenced the adjacent waters of the cays, the most recent taking place in 1961 (Hattie), 1971 (Irene), 1988 (Joan), 1996 (Cesar), 1998 (Mitch), 1999 (Lenny), 2007 (Felix), and 2010 (Matthew), according to the United States National Hurricane Center [15]. More recently, Hurricanes ETA and IOTA swept the archipelago, which produced vast devastation in the populated areas. The coral reefs have not been recently evaluated.

The estimates are tens of billion dollars as a result of increases in the hurricane frequency and the intensity in the Caribbean [16]. According to the Colombian Third National Communication on Climate Change, the San Andres Archipelago possesses the

greatest vulnerability in Colombia, having the highest risk values because of the threat that is perpetrated by hurricanes [14].

The San Andres Archipelago has a population of 61,280, according to the 2018 census. Most people rely on the fisheries that have been created in the cays, including Alburquerque, which is just a few miles south of San Andres Island, although the sustainability of the coral-reef health is of the utmost necessity. Hurricanes arrive frequently, and the recent hurricanes ETA and IOTA affected the reef severely.

Moreover, the wave regime is changing. A climatic analysis of the waves of Alburquerque Island, based on NCEP–NCAR reanalysis made with data from 1948 to 2008, assessed the wave climatology in terms of the Hs12 and the mean flux energy direction. The authors of [17] permitted the extrapolation of its behavior, and they show that the wave-energy mean-flux direction is changing anticlockwise, with tendencies of 1.91° in 2025, 5.75° in 2055, and 9.59° in 2085 [10]. Such changes may have important effects on the reef sustainability.

Modeling the wave regime and the formation of the water circulation in the coral environment of the cays of Alburquerque Island was the objective of the present study. This document aims to understand the reef ocean dynamics and its direct relationship to the functioning of the ecosystem, since the significant energies that are involved should play a fundamental role in the coral and fish distributions in the coral reef.

The following section presents the hydrodynamic model that was used for the case study. In Section 3.1, the wave climate is analyzed by using the results of a previous reanalysis, and Section 3.2 presents the results of the wave and current modeling.

## 2. Materials and Methods

Let us consider that, in the dynamics of the study area, the induced currents in the waves play the greatest role and that the patterns of the induced circulation are stationary. To generalize the mathematical formulation, we introduce the orthogonal curvilinear coordinate system in unstructured meshes. By looking for the time average of the shallow-water equations in a 2-D approximation with rigid-lid surface conditions and by filtering the gravitational waves, the system of the resulting governing equations is as follows:

$$\left[ \frac{\partial(\mathbf{J} < U^1 > H)}{\partial \xi} + \frac{\partial(\mathbf{J} < U^2 > H)}{\partial \chi} \right] = \alpha, \tag{1}$$

$$\frac{\partial(\gamma_1 < U^1 >^2)}{\partial \xi} + \frac{\partial(\gamma_{12} < U^1 >< U^2 >)}{\partial \chi} + \gamma_{1j} < U^j > \Gamma^1_{kj} < U^k > -f < U^2 >=$$
$$-g\mathbf{g}^{11} \frac{\partial < \eta >}{\partial \xi} - < F^1 > + \frac{< T^1 >}{\rho H} + < M^1 > +N^1_m \tag{2}$$

$$\frac{\partial(\gamma_{12} < U^1 >< U^2 >)}{\partial \xi} + \frac{\partial(\gamma_2 < U^2 >^2)}{\partial \chi} + \gamma_{2j} < U^j > \Gamma^2_{kj} < U^k > +f < U^1 >=$$
$$-g\mathbf{g}^{22} \frac{\partial < \eta >}{\partial \chi} - < F^2 > + \frac{< T^2 >}{\rho H} + < M^2 > +N^2_m \tag{3}$$

In a two-dimensional $\Omega$ domain with a sufficiently smooth boundary $\partial\Omega$ in the curvilinear coordinate system $(\xi,\chi)$, a Jacobian **I** of nonzero, and limited transformation, we obtain:

$$\mathbf{I} = \frac{(\xi, \chi)}{(x, y)}, \mathbf{J} = \mathbf{I}^{-1}. \tag{4}$$

The components $U^i = \mathbf{V}e^i$ in Equations (1)–(3) are contravariant of the vector of the currents (**V**) with the basic contravariant vector $(e^i = \zeta^i,)$, where $\zeta^i = (\xi,\chi)$, and i = 1, 2 (the contravariant components of the tensor metric $\mathbf{g}^{ik} = e^i e^k$; and $\Gamma^i_{kj}$, which are the Christoffel symbols of type II). Here, the chosen coordinate system is orthogonal; that is, $\mathbf{g}^{ik} = 0$, i ≠ k; i, k = 1, 2 (the coordinates are Cartesian when $\mathbf{g}^{11} = \mathbf{g}^{22} = 1$).

The operation < . . . > in Equations (1)–(3) means that the temporal average is on a scale that is greater than the period of wind and tidal waves, and it applies to both the

components ($U^i$ ($\xi,\chi$)) and the sea level ($\eta$ ($\xi,\chi$)). The other symbols are as follows: g: gravity; $\rho$: density of water; h, H: local and total depths, respectively (H = h + $\eta$); f: Coriolis parameter; $T^1$ and $T^2$: the respective components of $\tau_{sx}$ and $\tau_{sy}$ in the Cartesian plane (x,y) of the wind stress in the curvilinear coordinates; $\gamma_U$, $\gamma_{UV}$, and $\gamma_V$: expressions that parameterize the vertical structure of the flow [18].

The term $M^i$ was specifically introduced in Equations (2) and (3) to describe wave currents. These are radiative stress components that are produced by waves (wave-induced force per unit surface area (gradient of radiation stresses)) [19].

The $F^i$ term in Equations (2) and (3) is related to the bottom friction effects. In the case of wave dynamics, these terms are presented as recommended in [20]. For the effect of currents, the quadratic law of friction applies. In the coupled form of these two effects, the formulas are linearized by employing an integral coefficient of the bottom friction (r). Thus, $F^i = rU^i$.

It is assumed that the variations in the sea level and the flows have been filtered, and, for this reason, Equation (1)—which is in curvilinear coordinates in terms of the average current—is presented in a quasi-divergent form (with an expansion of $\alpha$). Equations (2) and (3) include the additional terms $N^i{}_m$ (similar to $\alpha$ in Equation (1)), which do not have any physical sense (e.g., in the analogous comparison to Reynolds stresses), but are the products of mathematical operations. However, according to [21], these terms are tidal stresses that express the contribution of long tidal waves to the residual circulation:

$$N^1_m = -\left\langle \frac{\partial(\gamma_1 U^1_m{}^2)}{\partial\xi} + \frac{\partial(\gamma_{12} U^1_m U^2_m)}{\partial\chi} + \gamma_{1j} U^j_m \Gamma^1_{kj} U^k_m \right\rangle$$

$$N^2_m = -\left\langle \frac{\partial(\gamma_{12} U^1_m U^2_m)}{\partial\xi} + \frac{\partial(\gamma_2 U^2_m{}^2)}{\partial\chi} + \gamma_{1j} U^j_m \Gamma^1_{kj} U^k_m \right\rangle \qquad (5)$$

$$\alpha = -\left\langle \frac{\partial(J\eta_m U^1_m)}{\partial\xi} + \frac{\partial(J\eta_m U^2_m)}{\partial\chi} \right\rangle.$$

The subscript "m" in Equation (5) then represents the instantaneous-velocity components of the tide with the respective sea level.

To find the in-plane scalar vortex function, the covariant components gii = 1/gii are used. Taking advantage of the fact that Equation (1) is nondivergent in the (x,y) plane, the definition of the stream function ($\psi$) is introduced according to the following relationships:

$$J\left[\left\langle U^1 \right\rangle H + \left\langle U^1_m \eta_m \right\rangle\right] = -\frac{\partial\Psi}{\partial\chi} \; ; J\left[\left\langle U^2 \right\rangle H + \left\langle U^2_m \eta_m \right\rangle\right] = \frac{\partial\Psi}{\partial\xi}. \qquad (6)$$

Equations (5) and (6) include the tidal components. Nevertheless, under microtidal conditions with tidal currents of O (1–10) cm/s, the residual current velocities that are used in Equations (5) and (6) are less than 5–10% of the tidal currents. Thus, this effect is negligible, and it is analog to the wind-driven or thermohaline currents close to the barrier reef, where other physics should predominate, such as wave-induced currents. According to Equation (6), Continuity Equation (1) holds automatically. The rotation operation, which is applied to Equations (2) and (3) and multiplied by $g_{ii}$, produces the vortex equation in terms of the stream function:

$$\frac{\partial}{\partial\xi}\left(\frac{g_{22} rJ^{-1}}{H}\frac{\partial\Psi}{\partial\xi}\right) + \frac{\partial}{\partial\chi}\left(\frac{g_{11} rJ^{-1}}{H}\frac{\partial\Psi}{\partial\chi}\right) - \frac{\partial}{\partial\xi}\left(\frac{g_{22} fJ^{-1}}{H}\frac{\partial\Psi}{\partial\xi}\right) + \frac{\partial}{\partial\chi}\left(\frac{g_{11} fJ^{-1}}{H}\frac{\partial\Psi}{\partial\chi}\right)$$

$$= \frac{\partial R(\chi)}{\partial\xi} - \frac{\partial R(\xi)}{\partial\chi} \qquad (7)$$

where the right-hand side is the following rotational equation:

$$\mathrm{rot}R(\xi,\chi) \equiv \frac{1}{\rho}\left[\frac{\partial}{\partial\xi}\left(g_{22}\frac{<T^2>}{H}\right) - \frac{\partial}{\partial\chi}\left(g_{11}\frac{<T^1>}{H}\right)\right] +$$

$$+ \frac{\partial}{\partial\xi}\left[g_{22}\left(\hat{N}^2_m + <M^2> - L(U^2)\right)\right] - \frac{\partial}{\partial\chi}\left[g_{11}\left(\hat{N}^1_m + <M^1> - L(U^1)\right)\right] \qquad (8)$$

The term $\hat{N}_m^i$ now also includes the correlations between the velocity and the instantaneous level:

$$\hat{N}_m^1 = N_m^1 - \frac{1}{H}\left[f < U_m^2 \eta > + r < U_m^1 \eta > \right]; \tag{9}$$

$$\hat{N}_m^2 = N_m^2 - \frac{1}{H}\left[-f < U_m^1 \eta > + r < U_m^2 \eta > \right], \tag{10}$$

The quasi-nonlinear terms $L(U^i)$ can then be presented in the following way:

$$L(U^1) = \frac{\partial(\gamma_1 < U^1 >^2)}{\partial \xi} + \frac{\partial(\gamma_{12} < U^1 >< U^2 >)}{\partial \chi} + \gamma_{1j} < U^j > \Gamma_{kj}^1 < U^k >$$

$$L(U^2) = \frac{\partial(\gamma_{12} < U^1 >< U^2 >)}{\partial \xi} + \frac{\partial(\gamma_2 < U^2 >^2)}{\partial \chi} + \gamma_{2j} < U^j > \Gamma_{kj}^2 < U^k >$$

In the solid boundary ($\partial\Omega_k$), the impermeability conditions for the water flow were adjusted such that the flow that was perpendicular to the boundary was equal to zero. In terms of the stream function, this condition will be:

$$\psi = Const_k, \text{ in } \partial\Omega_k, k = 1, \ldots, N, \tag{11}$$

where N is the quantity of the continuous fragments of the solid boundary. If $\psi = 0$ is specified for one of the two cay islands, then the value $\psi = Const \neq 0$ in the other must be calculated by using Equation (6) in the iterative process of Equation (7).

At open boundaries, the flow conditions for the stream function could differ, depending on the physical assumptions that are made about the flow. In general, the use of a numerical extrapolation from the calculation domain to the liquid boundary is recommended:

$$\frac{\partial^j \psi}{\partial n^j} = 0 \tag{12}$$

where j is the order of the derivative. Usually, extrapolations of the order 0 (j = 1) or the order 1 (linear, j = 2) are used.

For the calculation domain, we analyzed the available bathymetric data that were obtained by the echo sounder from the Colombian Caribbean Center for Oceanographic and Hydrographic Research (CIOH) and the LIDAR data, as processed by the United States Naval Oceanographic Office (Navoceano). These data were acquired with a spatial resolution of 4 m in waters with a depth of less than 50 m. Very high-resolution data, which were obtained from Navoceano, to describe the coral reef dynamics and the large distances (various kilometers) between the atolls, as well as the deep waters for the wave-climate conditions, are circumstances in which to employ nested grids with different spatial resolutions (Table 1). The mesh-nest-formation process was completed by using the model in the three abovementioned calculation domains.

**Table 1.** Numerical data for three nested grids of the model.

| Domain Number | 1 | 2 | 3 |
|---|---|---|---|
| X min (UTM), m | 400,770 | 405,958 | 408,046 |
| X max (UTM), m | 413,500 | 412,043 | 410,565 |
| Y min (UTM), m | 1,338,735 | 1,341,635 | 1,343,687 |
| Y max (UTM), m | 1,350,185 | 1,349,260 | 1,345,650 |
| X length, m | 12,730 | 6085 | 2519 |
| Y length, m | 11,450 | 7625 | 1963 |
| Resolution, m | 50 | 20 | 8 |
| Grid points | 256 × 230 | 305 × 382 | 316 × 246 |

If the domain with the available echo-sounder data is approximately 20 × 20 km (Figure 3), the selection criterion of the first mesh (the coarsest) is taken as a depth of 200 m

for the deep-water limit. In this way, the first, second and third selected domains, as shown in Figures 3–5, are provided in the following UTM coordinates (Table 1).

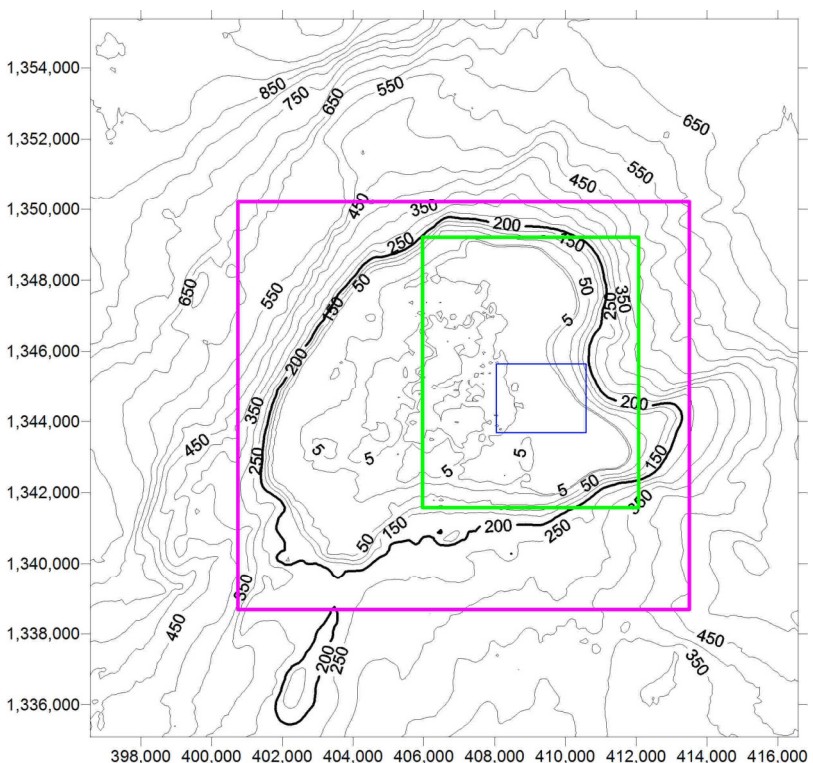

**Figure 3.** Bathymetry (in m) and three domains of interest: Mesh 1 (magenta polygon line); Mesh 2 (green); Mesh 3 (blue).

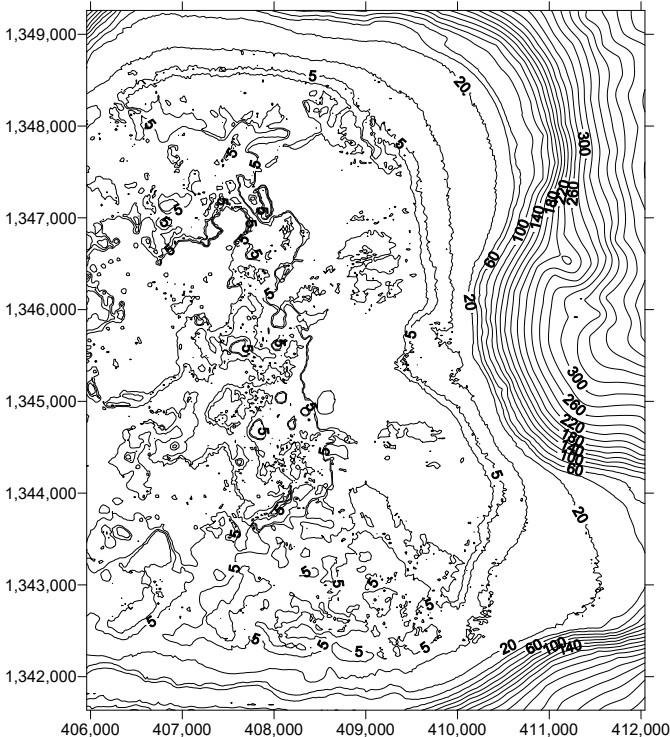

**Figure 4.** Bathymetry (in m) of Domain 2 (20 m resolution).

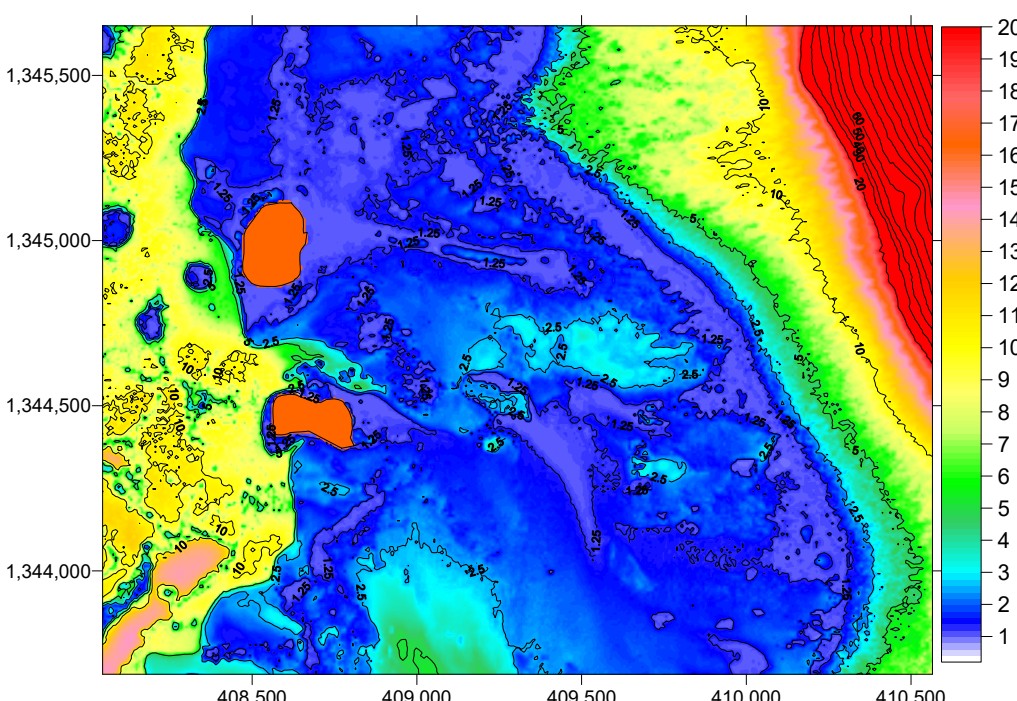

**Figure 5.** Bathymetry (in m) of Domain 3 (8 m resolution).

The wave climate was identified and applied in the deep waters in the contour of Domain 1 in the SWAN spectral wave model [22,23]. SWAN is a third-generation spectral wave model that is based on the wave-action-balance equation with sources and sinks. A 5-D wave spectrum, as an output, was used to provide the hydrodynamic model with the following information: the wave height and the wavelength, its mean and peak periods, the near-bed motion velocities for the bottom friction calculation, and the radiation stress tensor. The model configuration is as follows: The numerical scheme for the stationary mode was SORDUP [22]. The bottom friction was defined by Collins [24], and depth-induced wave-breaking and white-capping were set as the default [25]. The frequency range was specified between 0.04 and 0.5 Hz, with 51 frequencies, and the spectral angular resolution was 3°. Finally, the width of the directional distribution of the incident wave energy was set to 31.5°.

The propagation was carried out from the first mesh, in a nested form, and up to a mesh with an 8 m × 8 m spatial resolution. Here, the third-generation SWAN spectral model provides inputs for the hydrodynamic model, with the abovementioned information.

## 3. Results

### 3.1. Analysis of the Wave Climate in Deep Waters

The wave time series for the area of the Alburquerque Cay Islands corresponds to a virtual buoy (Figure 6) that is located at the coordinates 12.2262° N, 81.7969° W (UTM coordinates 413,320 m, 1,351,694 m).

The wave regime (height, period, and direction) corresponds to the modeled pseudo-data that were generated by the Colombian Maritime Directorate (DIMAR) of the CIOH, by using the SWAN numerical model and the winds of the North American Regional Reanalysis project (NARR) from January 1979 to December 2010. The model was calibrated and validated with information from DIMAR in situ wave buoys [26].

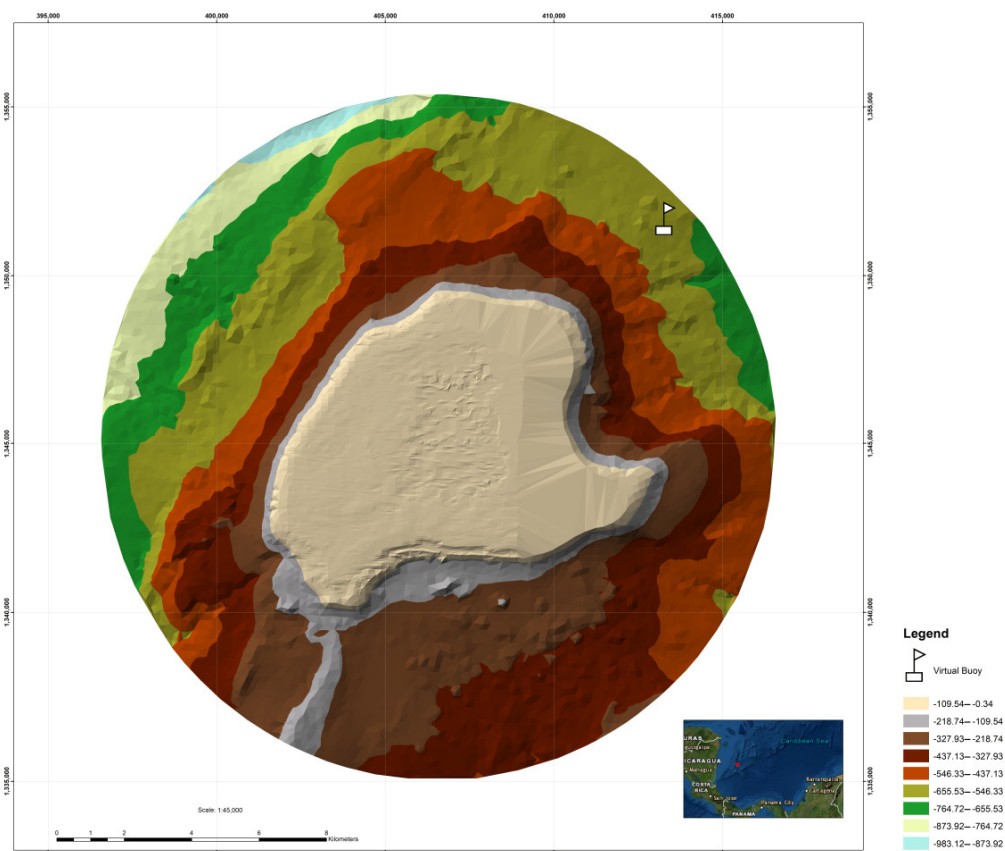

**Figure 6.** Location of the virtual buoy (white-flag symbol) for wave analysis.

The data on the significant height (Hs), the peak period (Tp), and the predominant direction were processed with CAROL software (Institute of Hydrology of the University of Cantabria), which thereby obtained the descriptors of the variables in the medium regime for the Hs and the Tp, and the extreme regime for the Hs.

The series of the Hs (Figure 7, top) has a mean of 1.14 m, with maximum and minimum values of 5.31 and 0.12 m, respectively. The standard deviation of the series is 0.54 m, with a mode of 0.79 m. The time series of the Ts (Figure 7, bottom) presents a mean of 7.44 s, a maximum of 20 s, a minimum value of 2.17 s, and a standard deviation of 1.43 s; the mode of the Ts is 7.72 s. In both series, the years 1988 and 1989 are missing. The significant height (Hs) is mainly distributed in the height bands that range from 0 to 2 m, as is seen in the histogram for the Hs (Figure 8).

The predominant direction of the mean swell is ENE, which includes more than 62.45% of the occurrences in the series for all of the quartiles, according to the wave increase (Figure 9) and the Hs statistics table (Table 2), with an $Hs_{99\%}$ of 2.48 m. The NE and E directions present probabilities of 13.21 and 8.7%, respectively, which indicates that these three directions contain 84.36% of the significant heights of the series. For the remaining directions, there is a 15.64% probability. The highest magnitude for $Hs_{99\%}$ is 4.53 m in the N direction, with a 2.13% probability.

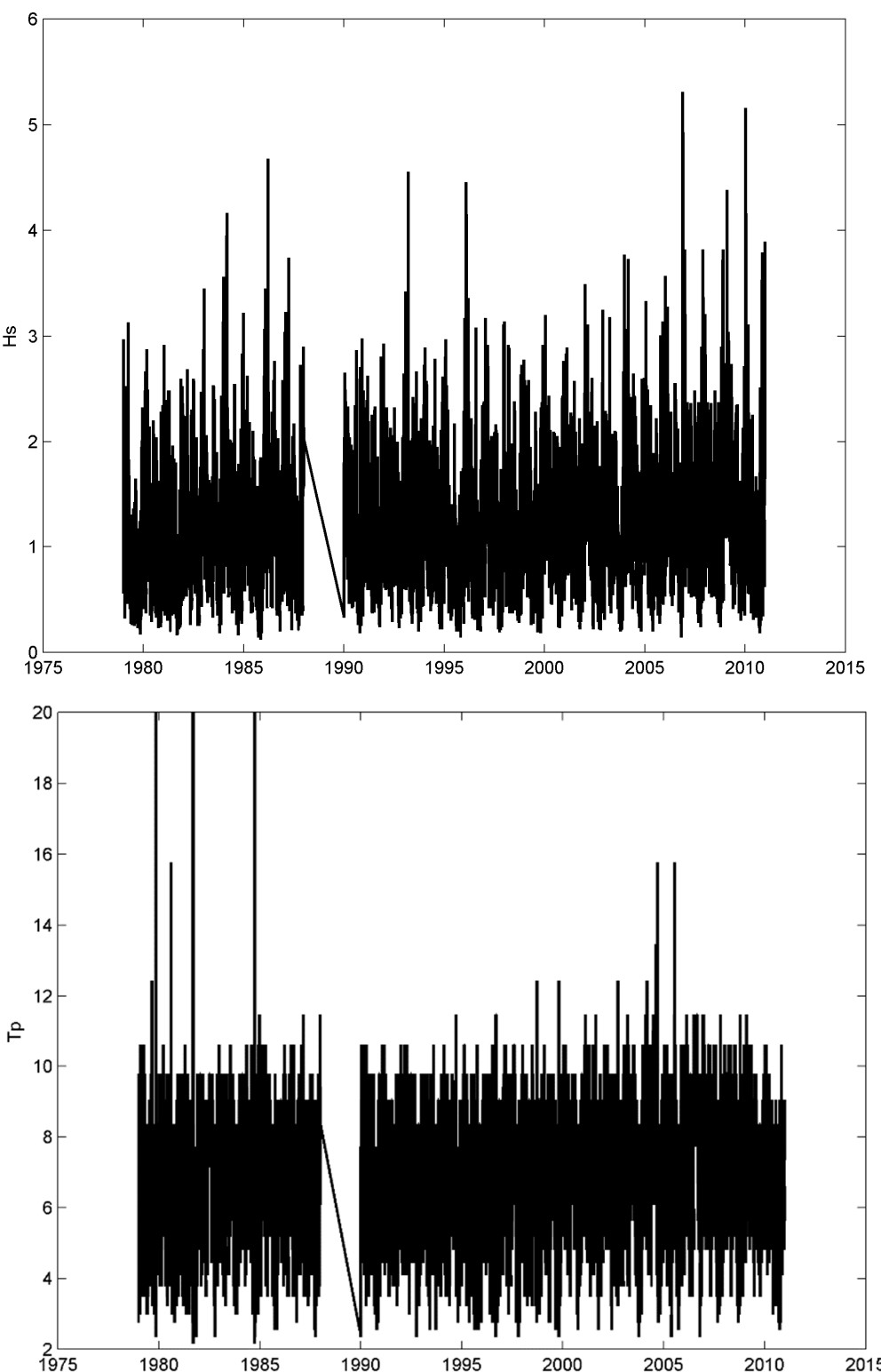

**Figure 7.** Calculated time series of the significant height (**top**) and peak periods (**bottom**) from January 1979 to December 2010. The years 1988 and 1989 are missing (gaps in these figures).

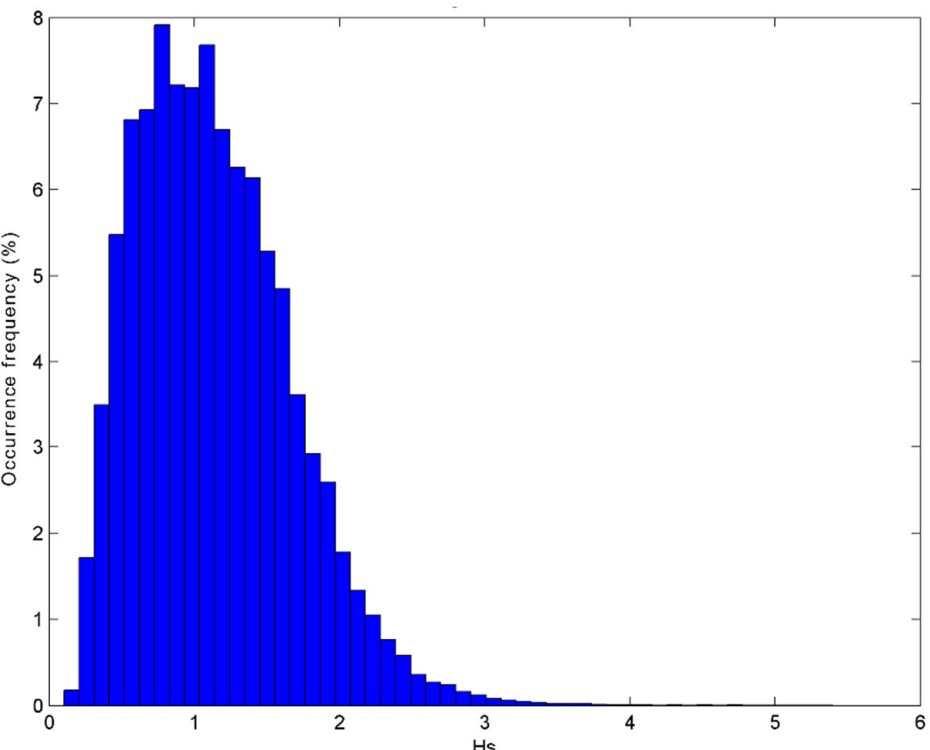

**Figure 8.** Histogram showing that the frequency of the significant height (Hs) is mainly distributed in the height bands that range from 0 to 2 m.

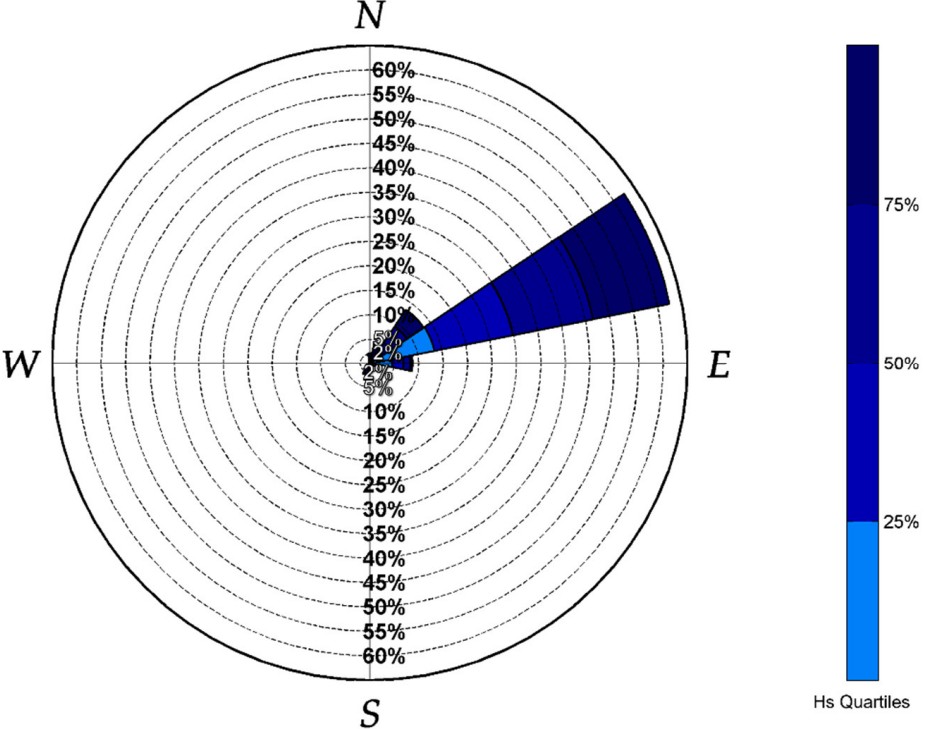

**Figure 9.** The probabilities for the wave height, the Hs, and the direction show that the ENE, NE, and E directions present a probability of 84.36% of the significant heights of the series.

**Table 2.** Statistics of the Hs (Hs$_{12}$ is a significant wave height that exceeded 12 h per year).

| Direction | Dir. Probability | Hs$_{50\%}$ | Hs$_{90\%}$ | Hs$_{99\%}$ | Hs$_{12}$ |
|---|---|---|---|---|---|
| N | 0.0213 | 1.4000 | 2.8700 | 4.5374 | 5.2143 |
| NNE | 0.0448 | 1.4200 | 2.2900 | 3.4919 | 4.6730 |
| NE | 0.1321 | 1.2400 | 1.8800 | 2.5868 | 3.1451 |
| ENE | 0.6245 | 1.1200 | 1.8400 | 2.4800 | 2.9100 |
| E | 0.0870 | 0.7100 | 1.3200 | 1.9120 | 2.2143 |
| ESE | 0.0167 | 0.5600 | 0.9670 | 1.5728 | 2.0500 |
| SE | 0.0081 | 0.6000 | 1.0500 | 1.5946 | 2.0342 |
| SSE | 0.0057 | 0.6000 | 0.9900 | 1.7577 | 2.0264 |
| S | 0.0073 | 0.6500 | 1.1660 | 2.0380 | 2.2734 |
| SSW | 0.0260 | 0.9300 | 1.5900 | 2.1348 | 2.9715 |
| SW | 0.0088 | 0.9500 | 1.6770 | 2.3482 | 2.5003 |
| WSW | 0.0072 | 0.9700 | 2.2600 | 3.5935 | 4.1352 |
| W | 0.0033 | 1.0250 | 2.5970 | 3.9160 | 4.4500 |
| WNW | 0.0014 | 1.0300 | 2.1120 | 2.6481 | 2.6700 |
| NW | 0.0012 | 0.8300 | 2.3570 | 2.7298 | 2.8400 |
| NNW | 0.0045 | 1.2400 | 2.1820 | 3.6626 | 4.8162 |

For the ENE direction (which was previously established as predominant), the Tp$_{99\%}$ is 10.6 s. The highest values of Tp$_{99\%}$ belong to WNW (11.74 s) and SSE (11.56 s), with probabilities of 0.14 and 0.57%, respectively, which likely correspond to occasional bottom swells within the series (Table 3).

**Table 3.** Statistics of the T$_p$.

| Direction | Dir. Probability | Tp$_{50\%}$ | Tp$_{90\%}$ | Tp$_{99\%}$ | Tp$_{12}$ |
|---|---|---|---|---|---|
| N | 0.0213 | 6.0800 | 7.7200 | 9.0500 | 9.7900 |
| NNE | 0.0448 | 6.0800 | 7.7200 | 9.0500 | 9.7900 |
| NE | 0.1321 | 7.1300 | 8.3600 | 9.7900 | 10.6000 |
| ENE | 0.6245 | 7.7200 | 9.0500 | 10.6000 | 11.4800 |
| E | 0.0870 | 7.1300 | 9.0500 | 9.7900 | 11.4800 |
| ESE | 0.0167 | 6.5900 | 7.7200 | 9.6716 | 10.6000 |
| SE | 0.0081 | 6.0800 | 7.7200 | 9.7900 | 20.0000 |
| SSE | 0.0057 | 4.8000 | 7.1300 | 11.5646 | 20.0000 |
| S | 0.0073 | 6.0800 | 7.7200 | 9.0500 | 11.1461 |
| SSW | 0.0260 | 7.7200 | 9.0500 | 9.7900 | 10.6000 |
| SW | 0.0088 | 5.1900 | 7.7200 | 9.0500 | 13.4500 |
| WSW | 0.0072 | 5.1900 | 7.1300 | 9.0500 | 9.0500 |
| W | 0.0033 | 5.6200 | 7.7200 | 9.0500 | 9.0500 |
| WNW | 0.0014 | 5.1900 | 7.1300 | 11.7432 | 12.4200 |
| NW | 0.0012 | 4.8000 | 7.7200 | 9.7900 | 9.7900 |
| NNW | 0.0045 | 6.0800 | 7.7200 | 8.3600 | 9.7626 |

*3.2. Wave Propagation and Wave-Induced Currents*

The reanalysis wave-climate pseudo-data were applied to the closest WW-III node (the virtual buoy in Figure 6) in deep water as the boundary conditions in the SWAN spectral model. The following figures correspond to the application of the wave and hydrodynamic model, according to the main wave patterns. Figures 7–9 correspond to the wave conditions in the three nested meshes. Ultimately, of all the directions that were recorded by reanalysis, the wave climate in this region was characterized by waves in the ENE sector (more than 62% of cases; Figure 9 and Tables 2 and 3), with a wave regime of 1.12 m, and featuring significant height and a period of 7.7 s. In this model, the intensity of the trade winds increases the swell to 2.9 m and 11.5 s, which preserves this direction.

Therefore, the approach for modeling the circulation of wave-induced currents was determined on the basis of this wave pattern, which propagates towards the islands, and which is considered the most conducive environment for coral-reef development. For these islands, this environment has an orientation that is strongly related to the average wave climate.

The following figures correspond to the application of the wave and hydrodynamic models, according to the wave climate. Figures 10–12 correspond to the wave conditions in the three nested meshes for waves with the highest levels of occurrence and energy. The barrier effect, moreover, is observed in the deformation of the wave fields, where the barrier attenuates the wave height. The breaking result by depth corresponds more strongly to longer waves. The shorter waves in the spectral packet refract from the jagged coral mounds along this barrier, which generates jagged rays that appear within the atoll in Figure 12 (top) in the wave height, while, in the wavelength (Figure 12, below), this phenomenon is not as noticeable.

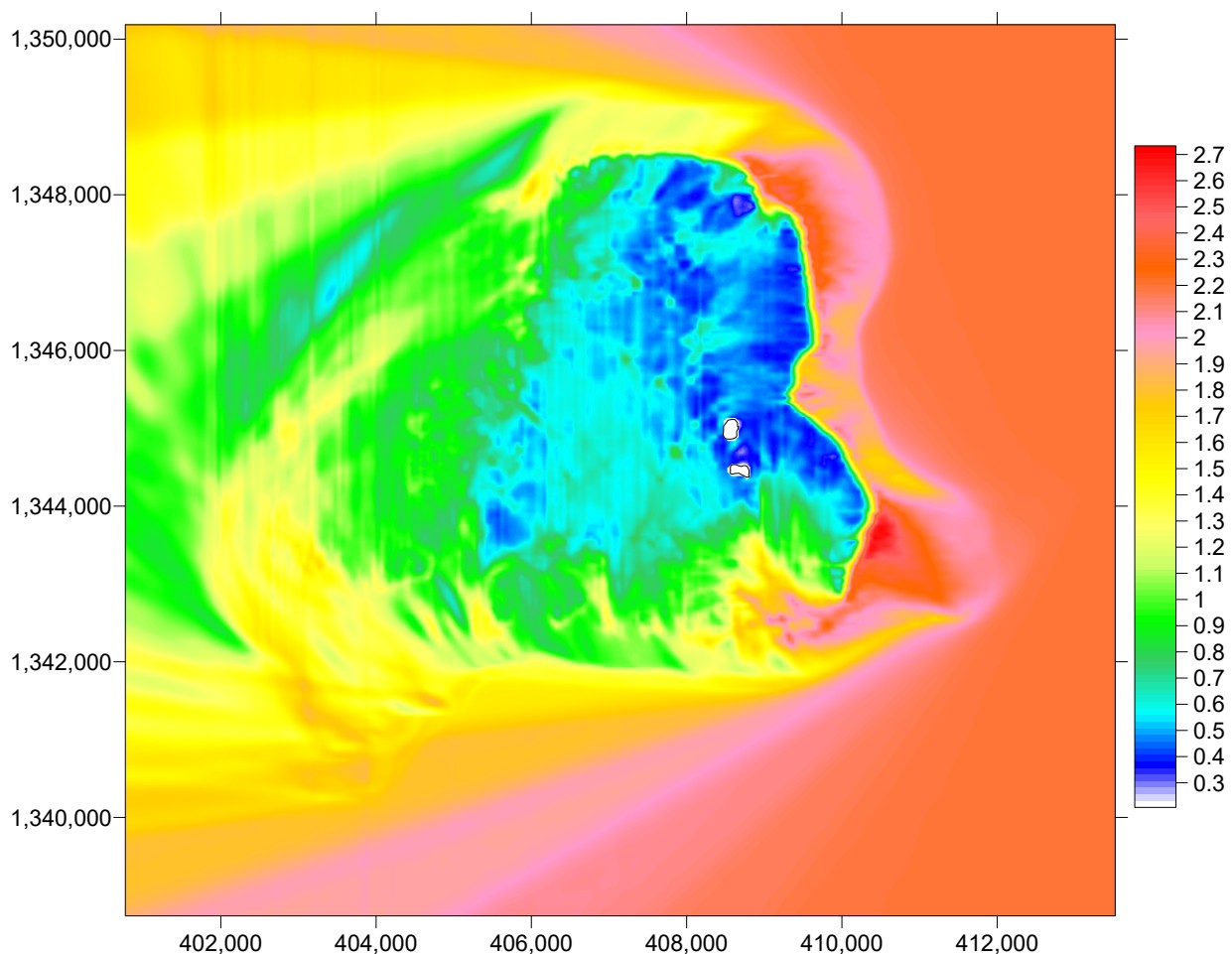

**Figure 10.** Significant wave-height field (in m) in Mesh 1 that corresponds to the propagation of the virtual buoy in deep water. Waves from the E. The barrier effect, moreover, is observed in the deformation of the wave fields.

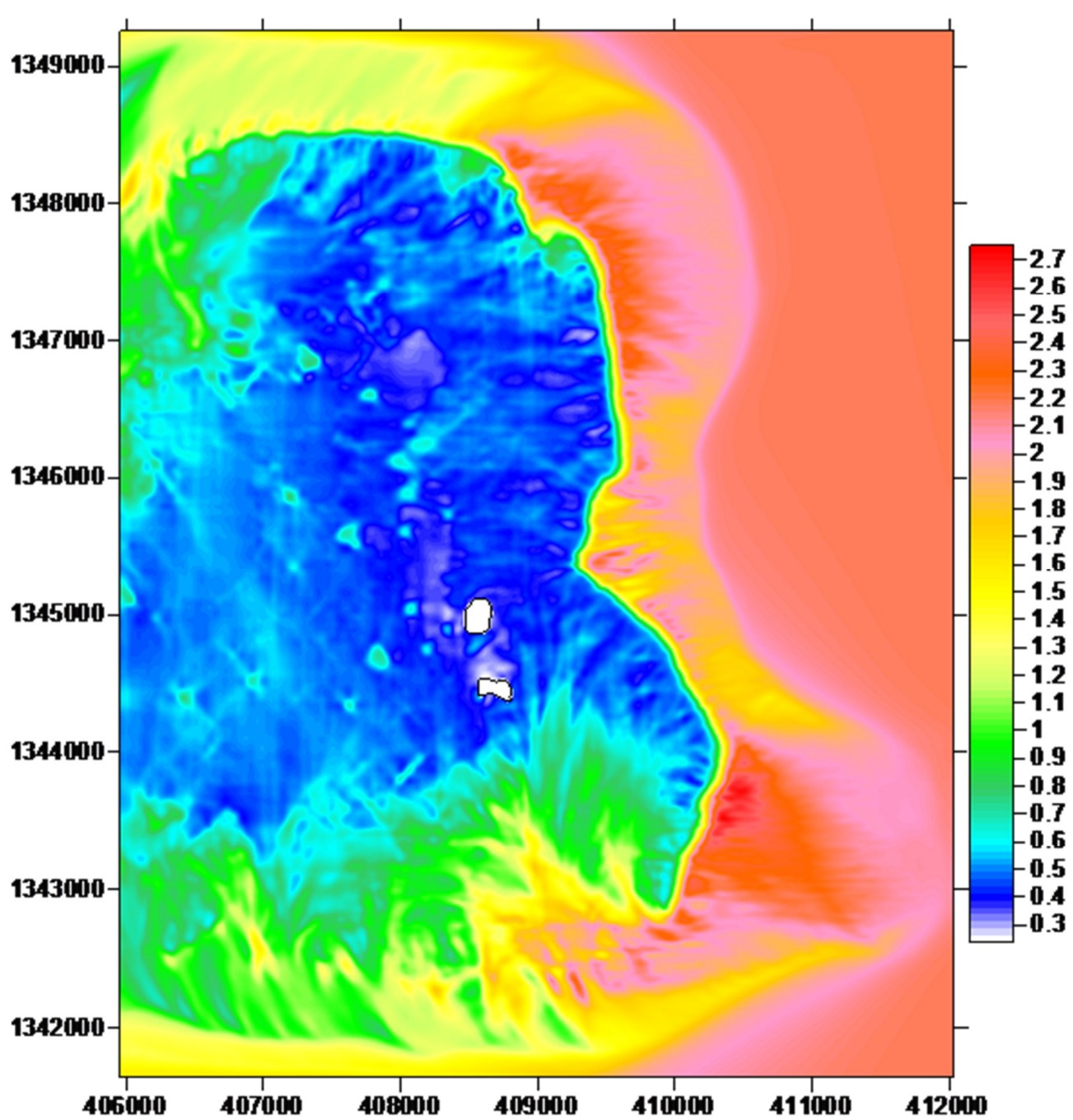

**Figure 11.** Significant wave-height field (in m) in Mesh 2 that corresponds to the propagation of the virtual buoy in deep water. Waves are from the E-generating jagged rays that appear within the atoll.

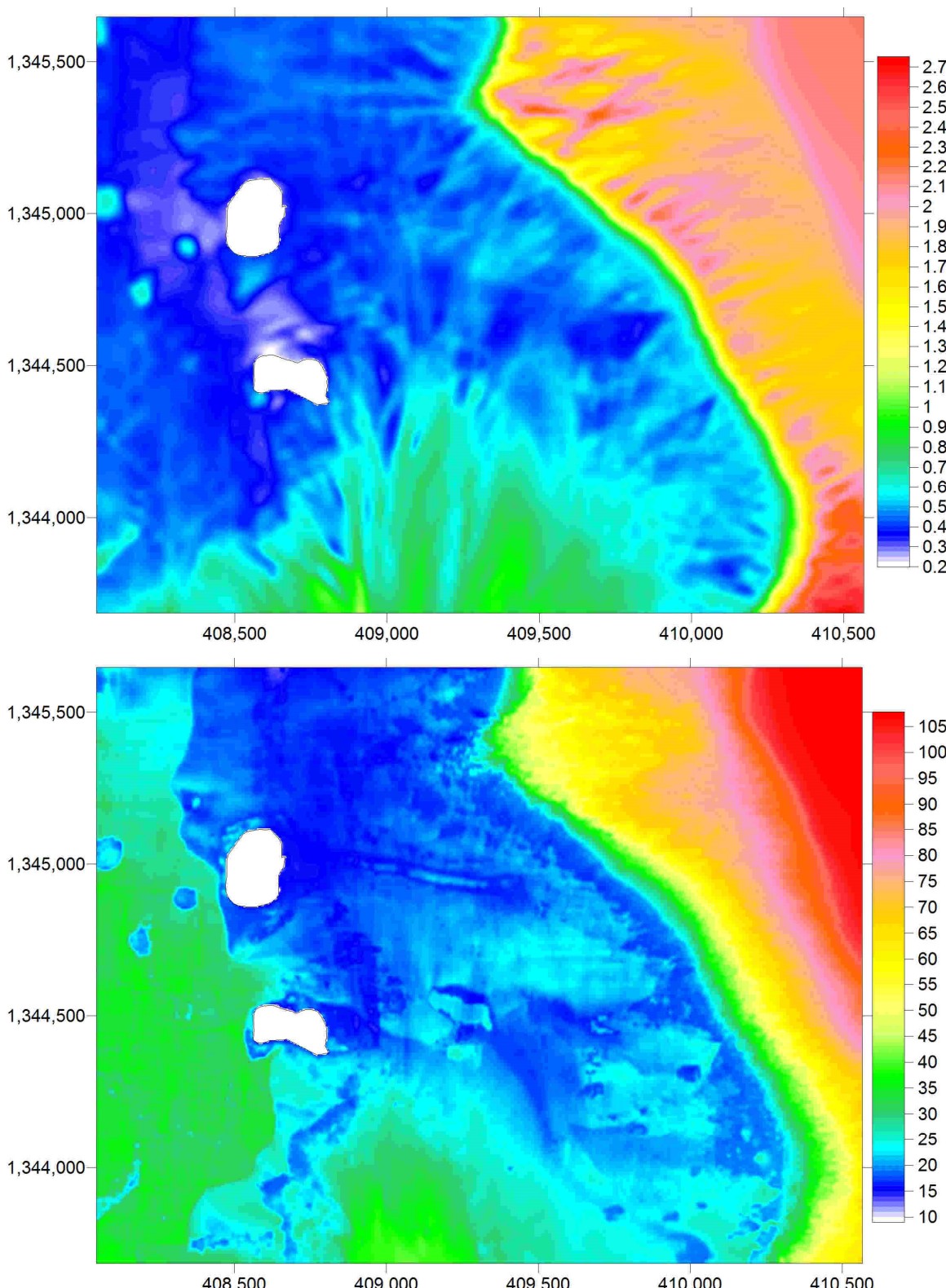

**Figure 12.** Significant wave-height field (**top**) showing jagged rays that appear within the atoll, which are not as noticeable in the wavelength field (**bottom**) (in m) in Mesh 3, and which correspond to the propagation of the virtual buoy in deep water. Waves from the ENE, Case Hs$_{12}$ (Tables 2 and 3).

In other words, the wave spectrum transformed at the barrier, with a decrease in waves of small wave numbers, and in the distribution of their energy to larger wave numbers with greater directional spreading.

The following analysis focuses on the effect of the wave transformation in the field of coastal circulation, the principal cause of which is, in this case, waves breaking over the coral barrier. Figure 13 corresponds to the northeastern sector of the domain (Figure 5), which shows the field of the wave steepness (*relationship between wave height and wavelength*) on the bottom relief. The presented case corresponds to a significant wave of 12 h during the year in order to analyze the waves over a long enough timeframe (with a period of 11.5 s) around two relevant slopes in the area: (a) a steep slope of the continental platform (20 to 100 m deep); and (b) a less steep slope with shallow depths in the barrier reef. A subsequent analysis was performed along the A–B transect (in Figure 13), which indicated that the values with the highest steepness are on the reef.

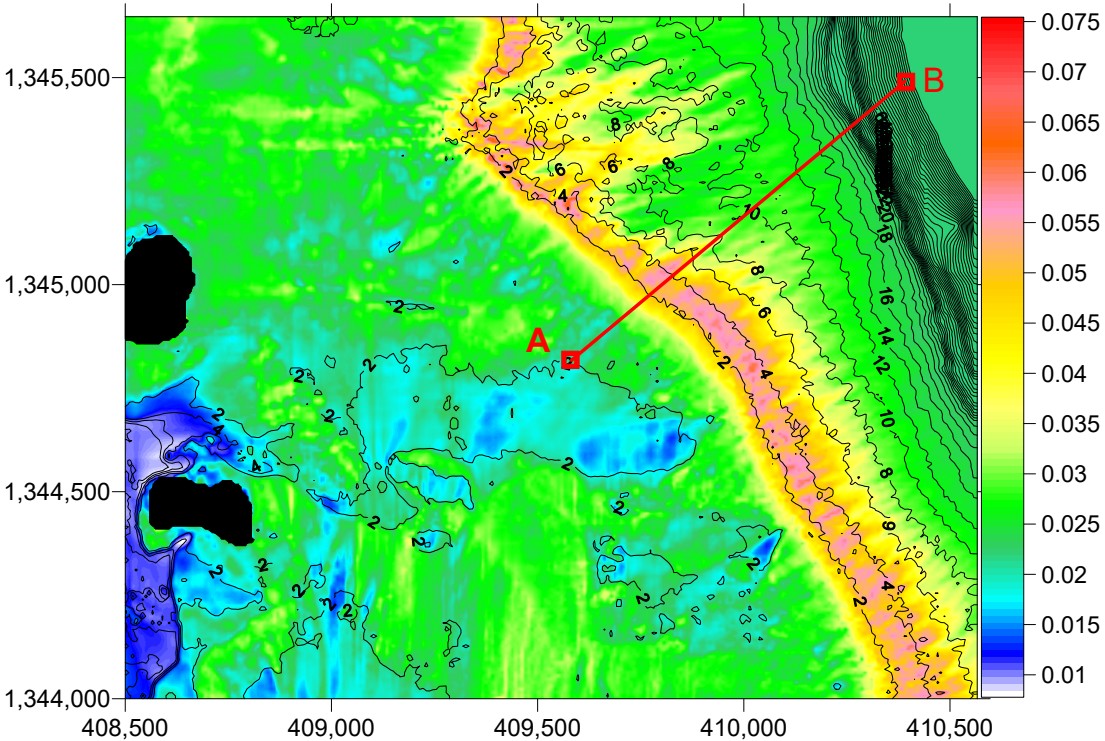

**Figure 13.** Wave steepness (magnitude in colors) on the bottom relief (bathymetry contours in meters). Transect A and B indicates the position used for analysis. Waves from the ENE, Case Hs$_{12}$.

Figure 14 presents one of the outputs of the hydrodynamic model for relatively long waves, which corresponds to the case outlined above. The formation of a jet can be observed along the barrier, with a flow velocity greater than 1 m/s. The isolines represent the stream function, whose magnitude in the jet varies here from 0 to 200 m$^3$/s, with a perpendicular distance of 50–100 m.

As shown in Figure 15, by varying the wave incidence angle in deep water (the virtual buoy, according to the data in Tables 2 and 3), this circulation pattern is conserved for the predominant wave, which slightly decreases the intensity of the induced circulation.

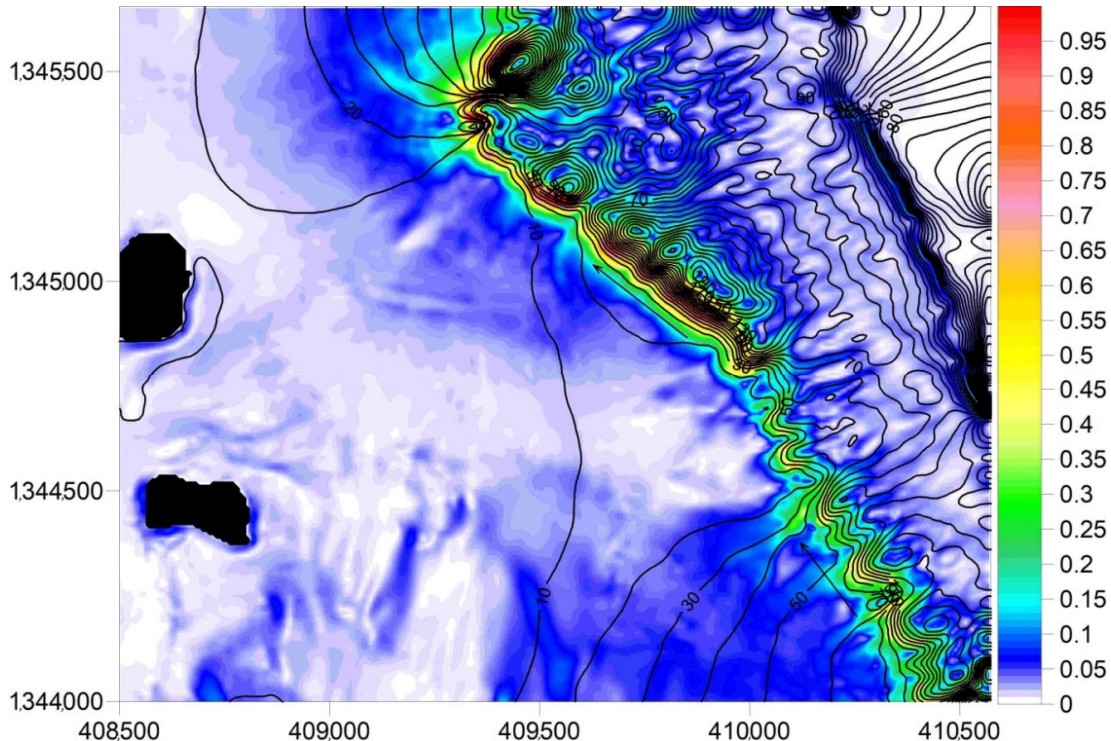

**Figure 14.** Stream function of currents (isolines in m$^3$/s) induced by waves corresponding to the E wave pattern and the Hs$_{12}$ swell. The maximum speed along the barrier reef is greater than 1 m/s (color scale in m/s). Arrows indicate the directions of the currents.

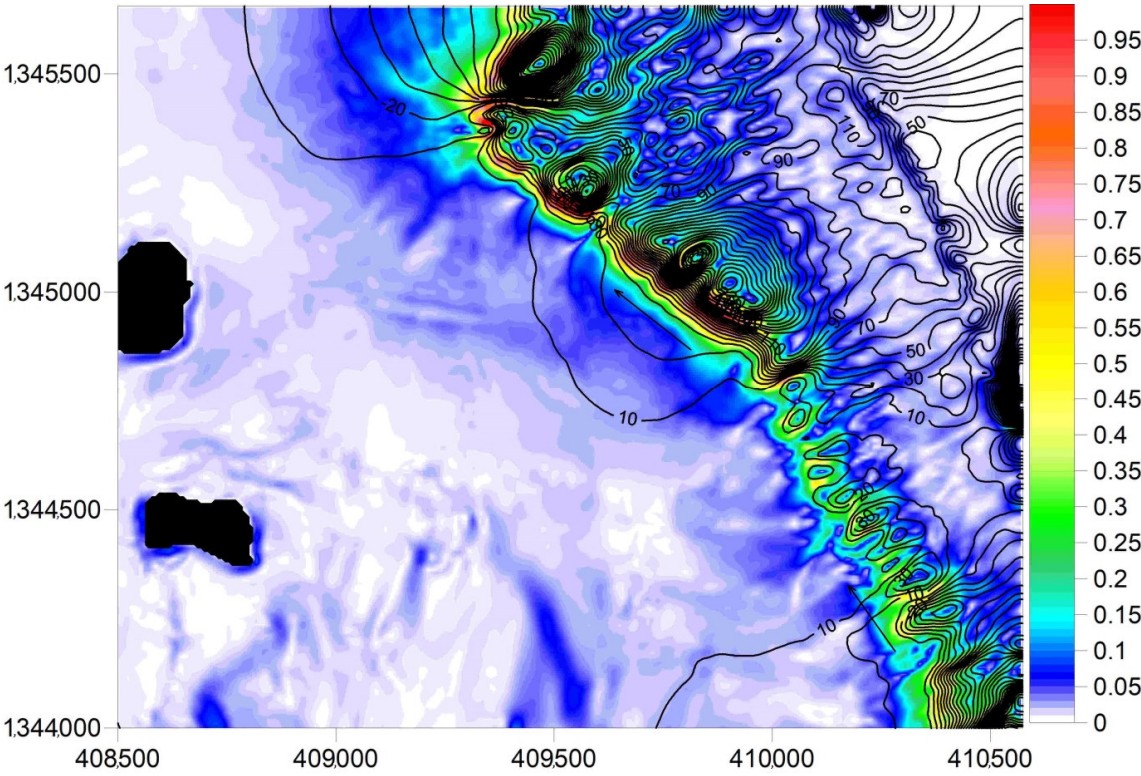

**Figure 15.** Stream function of currents (isolines in m$^3$/s) induced by waves that correspond to the ENE wave pattern for Wave Hs$_{12}$. The maximum speed along the barrier reef is greater than 1 m/s (color scale in m/s). Arrows indicate the directions of the currents.

However, when modeling the Hs waves of 50% (Tables 2 and 3) under the moderate regime with the average energy vector of the wave incidence, the disappearance of the water jet is observed (Figure 16), along with the formation of local recirculation systems around the coral mounds with nonuniform vertical elevations. Here, the eddy current velocity no longer exceeds 20–30 cm/s, and this represents the general abiotic conditions for the formation of the coral reef.

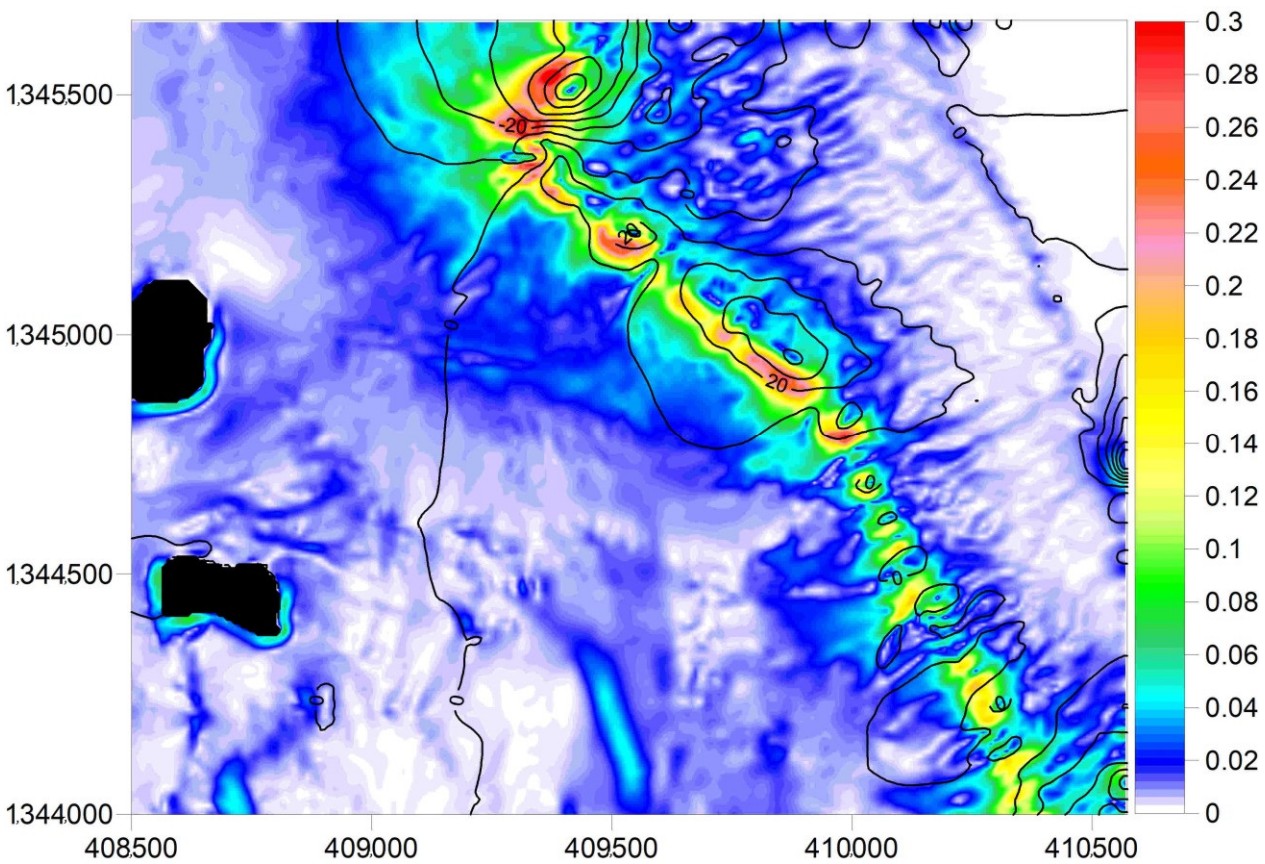

**Figure 16.** Stream function of currents (isolines in m³/s) induced by waves that correspond to the ENE wave pattern; waves of the regime (color scale in m/s).

The transects of the depth, the significant height, the wave-energy-dissipation rate, the bottom orbital velocity, the wavelength, and the steepness along Transect A–B (Figure 13) are shown in Figure 17. In this work, two cases were compared according to the climate: the wave of the 50% regime (*a median of Weibull distribution*), and the wave of 12 h during the year; both were from the same ENE direction. In both cases, trade winds are predominant in the region.

For a comparison to the results that were obtained and that are presented in Figures 14 and 16, we assumed the existence of a wave-height threshold at which the circulation appears along the coral reef. We also assumed that this threshold is above an Hs of 1.2 m, as is shown in Figure 17. In other words, the average case that is shown in Figure 16 reflects the circulation that is associated with the wave regime, and it is more usual and representative than the case in Figure 14.

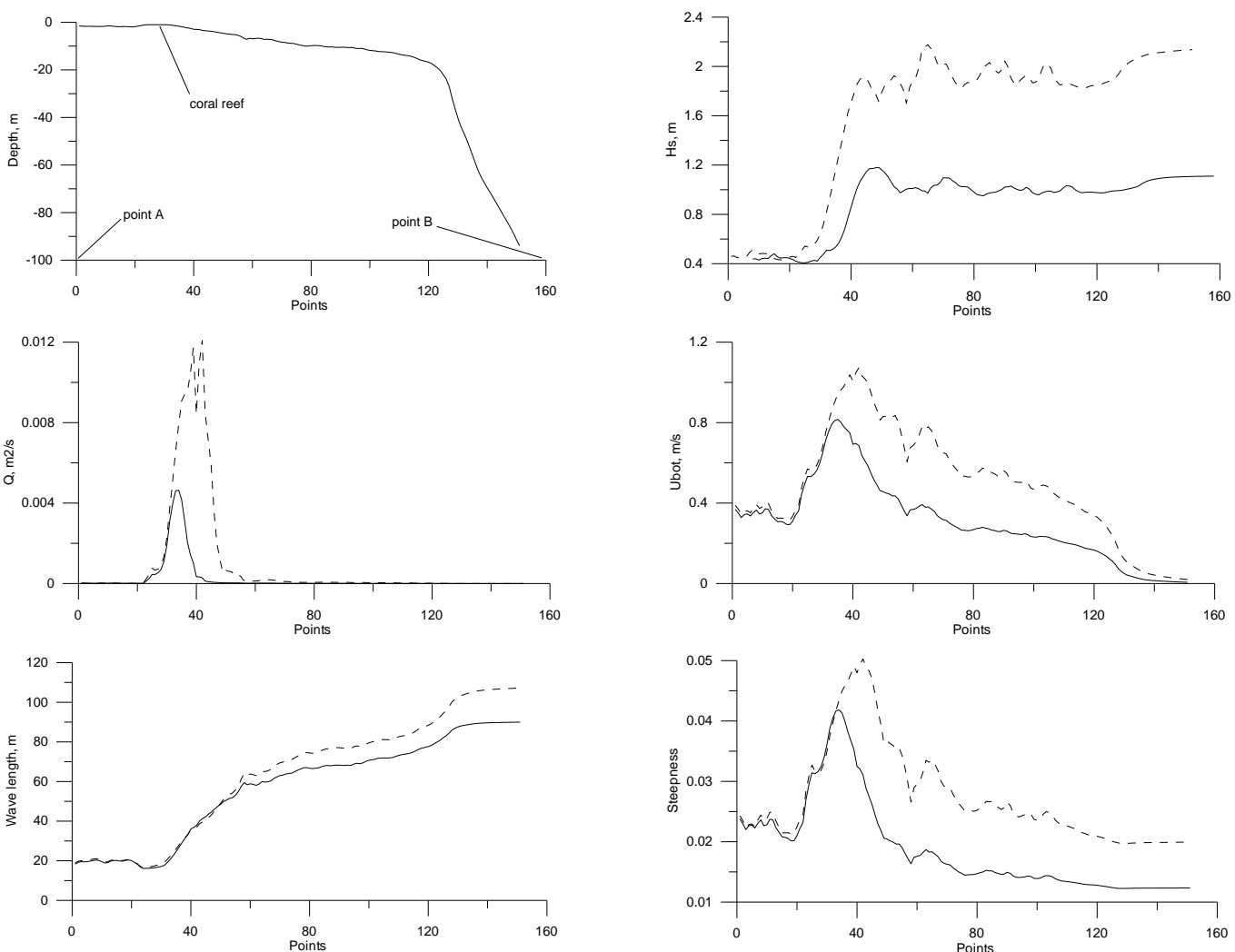

**Figure 17.** Profiles of depth, significant height (Hs), wave-energy-dissipation rate (Q), bottom orbital velocity (Uorb), length, and steepness for waves of the regime (solid line) and the $Hs_{12}$ (dashed line) along Transect A and B (in Figure 13).

The specific rate of the dissipation of the wave energy at the break between these two states varies from $4 \times 10^{-3}$ to $12 \times 10^{-3}$ m²/s (i.e., three times in energy terms), while the orbital speed at the bottom only increases from 0.8 to 1 m/s between the medium and extreme cases. The wavelength, in turn, does not show significant differences. Moreover, the wavelength's impact on the steepness value is small compared to the increase in the wave height along the underwater transect.

## 4. Discussion

Coral barriers play a fundamental role in protecting the atoll space against wave action, according to the same physics as soft substrates. These barriers form sand bars that protect the coastline from erosion during storm events. However, the temporal dynamics of coral reefs and their developments occur over very long timescales, and they are not comparable to hydrodynamic processes that are related to the passage of tropical cyclones and cold fronts; these events are considered to be extremes in the Seaflower region. Therefore, the adaptation of a reef to abiotic environments must correspond to multi-year average conditions, and not to extreme conditions.

The databases that were used in this paper did not clearly or significantly reflect the passage of cyclones or frontal systems. Indeed, a wave reanalysis based on a previous

atmospheric analysis was barely able to record the strong winds of a "squall" within the narrow strip of a cold front on the water. These winds were difficult to detect by using the spatial difference in the temperature and the wind direction. Since the purpose of this research was based on the assumed average wave regimes, the wave dynamics were associated with the influence of the trade winds. We determined that the average energy flow most likely comes from the ENE sector.

Clearly, the circulation over the islands is generally formed by the currents that are induced by waves as part of their mechanisms of breaking. Therefore, the research focus should be on this type of process, and it should omit wind-driven, thermohaline, and tidal currents. The applied hydrodynamic model indicated the bi-modal climate of water dynamics over the coral reef, which reveals that: (1) under the same wave-incidence angle, a circulatory system is formed (or not) along the barrier; and (2) there is a threshold for the wave height when slow circulation over the coral mounds locally transforms into a jet along the reef. Bi-modal circulation patterns should directly affect the coral larvae migration along the barrier or around the coral mounds, which produces more or less effects on the reef development. The existence of such a bi-modal form would be impossible on an impermeable coast, where changes to the regime depend on the angle of the incidence of the waves. This factor creates a difference between coral (semipermeable) and coastal environments.

The resolution of this particular case of circulation over coral reefs is fundamental for understanding the coral and fish distributions in these complex ecosystems that are the basis for the island's population. These simulations also permit the assessment of scenarios that follow variable patterns that will be due to climate change in the future. Variations in the wave-climate conditions that are due to climate change may alter the reef formation.

**Author Contributions:** S.L. designed and ran the circulation model; C.A.A. and J.M. contributed to the wind, wave, and oceanographic analyses, and to the writing—review and editing of the manuscript. All authors have read and agreed to the published version of the manuscript.

**Funding:** The paper is one of the results of the project "Characterization of the morphology of the seabed and its relationship with physical processes of the ocean", which was financed by the Colombian National Science Foundation "Francisco José de Caldas, Colombia (Project No. 65030/2019), and by the Colombian Naval Academy.

**Institutional Review Board Statement:** Not applicable.

**Informed Consent Statement:** Not applicable.

**Data Availability Statement:** Not applicable.

**Acknowledgments:** The authors express their gratitude to the Colombian Caribbean Center for Oceanographic and Hydrographic Research (CIOH) for providing the bathymetric data.

**Conflicts of Interest:** The authors declare no conflict of interest.

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
