# Peer review of "Wave Climate and the Effect of Induced Currents over the Barrier Reef of the Cays of Alburquerque Island, Colombia"

_sustainability, doi:10.3390/su14106069_

Round 1

Reviewer 1 Report

This is a review of the MPDI Sustainability Oceans [manuscript id 1631608] describing the “Wave Climate and the Effect of Induced Currents over the Barrier Reef at the Cays of Albuquerque Island, Colombia”.

– General review and comments –

Although the main objective of this article is clear: “to model the wave regime and the formation of water circulation in the coral environment”, there is a great doubt here as to how this knowledge contributes to one or more points defined in the Oceans section of Sustainability (see at ** below). I believe that the authors could better explore a few paragraphs in the introduction and conclusion about the importance of this type of study, especially if we are talking about environmental vulnerability. However, the article does not explore any of the important aspects of the area of ​​study within the main theme of the journal. In the same way, I felt a lack of indications about the occupation of these islands and their regional importance in the introduction. It is notable for example that this area is in dispute at the International Court of Justice (https://www.icj-cij.org/en/case/155) between Colombia and Nicaragua. Even without going into this merit, it is important that the authors indicate geographic characteristics that make this environment important, in addition to the coral barrier itself (is there any type of occupation on the islands? Is the surrounding coral barrier a protected park or is it commercially exploited by the fishing?) Note that these are not minor issues as this is a manuscript submitted to a journal that aims to “advance our understanding of all aspects of the environment, biology, energy, engineer, ecosystem functioning, and human interactions with the oceans” . Thus, the pure physics and mathematics of waves does not apply to the central scope of this journal.

link cited above ** https://www.mdpi.com/journal/sustainability/sections/Sustainable_Oceans

Likewise, the authors performed a good mathematical discretization of the problem in the Material and Methods section, but it seems to “do not connect” specifically with the object of study. For example, “The nest mesh formation process” is indicated on lines 147 and 148 for the first time, without however discretizing why nesting is necessary for this type of simulation. If figures 3, 4 and 5 were produced in an integrated way, and cited “before” this line, the need for simulation using nesting would become clearer. However, this link between nesting meshes and the theory above line 143 is confusing.

Another example of confusion is that a brief search on the word SWAN reveals that it was used only 5 times throughout the text (+1 time in the abstract and once in the keywords). That is, the authors do not describe what the SWAN model is, who built it, how it works and how it is widely used in the simulation of waves similar to this manuscript, limiting themselves to citing the classic reference by Booij & Holthuijsen (1993). There is a huge volume of articles and reference bibliography on the SWAN model and its importance, and I believe the correct reference to the model used would be Booij, N., L. H. Holthuijsen, and R. C. Ris. "The" SWAN" wave model for shallow water." Coastal Engineering 1996, 1997, 668-676. or at least Booij, N.R.R.C., Roeland C. Ris, and Leo H. Holthuijsen. "A third-generation wave model for coastal regions: 1. Model description and validation." Journal of geophysical research: Oceans 104.C4 (1999): 7649-7666. The authors MUST improve the theoretical framework regarding the use of the SWAN model for this type of simulation, in the same way that the references (only 10 !) are far below what is expected for a manuscript of this type.

It was quite complicated to extract information from simulated scenarios from the text. There is no table (for example, as a suggestion the authors should indicate in a table the boundary conditions of the two simulated cases to facilitate the reader's understanding), indicating which parameters were used in the two scenarios. The results section begins directly with the analysis of the wave climate in deep waters, based on “virtual” time series, without an explanation regarding the technical choices for this data. No in situ measurement? So this was the only way to simulate wave climatology? At least one sentence saying this is needed here.

Figure 6 is quite confusing (are those individually observed bathymetry points?) and as there is no grid overlap (compared to Figure 3, for example, which has another geographic scale pattern. Some figures are in UTM and others in Lat-Long) it is quite difficult to understand the process of choosing the position of this virtual buoy. The text still refers to a calibration with an in situ buoy (reference 10) which is not available on the internet, only a summary of this work. Therefore, there is no way to “validate” the reference used for “validation” as indicated by the authors. Likewise, the pattern shown in Figure 7 (Tp and Hs) have “gaps” and were not correctly graphed. Thus, item 3.1 of lines 174 to 217 should at least present a quality of wave statistics where it is possible to understand the choice of scenarios for simulation with SWAN. There are several references available in the literature for this type of comparison between wave modeling and climatology and authors MUST at least try to develop the manuscript in the same way.

Between lines 228 and 231, the authors finally “discuss” their choices for the wave-induced current simulation scenario, but too simply, with an abrupt transition to a series of figures (10-12) without delving into the discussion and interpretation of results.

The figures do not have a label on the color scale (the same color scale is used for Significant wave-height and wavelength field in figure 12) which is very confusing to the reader. In addition, these figures are at the scale of the model and not the map, without the proper geographical indications, which makes the reader have to look at other figures in order to follow the authors' reasoning.

The discussion on the two comparative cases with climatology (lines 292 to 312), including figure 17 here, should come in a separate item, together with the theoretical basis that led the authors to choose these scenarios. Likewise, figure 17 is well out of format and its visualization is hampered by the way it was produced. In this case, I suggest 2 figures, one with 2 frames side by side and the other with 4 frames, or lines superimposed on 2 figures with 2 frames each.

The authors also clarify that the simulations do not represent the passage of frontal systems. However, this declaration only appears on line 322, with no connection to the rest of the text. This explanation should be placed in the introduction. And moving on, the conclusion about the circulation (lines 330 to 339) doesn't seem clear as it stands, as the authors cite references 1 and 2 which are just numerical reviews on wave modeling (so it's a modeling manuscript only?).

Based on this assessment and the comments described above, I suggest that the manuscript be returned to the authors for a comprehensive and in-depth review, not only of the text, but also of the way in which the results are presented. Only then could it be recognized as a work to be published in Sustainability Oceans.

-- specific comments in some points of the manuscript —

  • There is no reference to tidal measurements or the tidal regime in the geographic description. The tide is only mentioned in the summary as a microtide, and obviously, in the mathematical discretization of materials and methods. Are there tidal measurements in the region or not? Precisely because from the mathematical point of view in eq. 6 the tides are important, they cannot be neglected at least in their description in the manuscript.
  • The domain information between lines 149 and 159 must be tabulated. The way it is written in the text is very confusing for the reader.
  • The introductory part, which contains the geographical and climatic description of the study area, from lines 23 to 79, has only 4 citations, while there are dozens (if not hundreds) of articles about the Archipelago of San Andrés region.
  • The statement in line 77 “This interrelation would be impossible without considering wave-induced currents as one of the main mechanisms of local circulation” must be supported by at least 2 other references with other environments that demonstrate similarity in the wave pattern.
  • It was also unclear why figures 1ABC appear in high resolution at the end of the text. This just demonstrates that the construction of figure 1 can be improved OR that some other detail is shown in these figures in high resolution.

Author Response

Author's Reply to the Review Report (Reviewer 1) – our answers in green (in the attached file)

Reviewer 2 Report

Lonin et al: Wave Climate and the Effect of Induced Currents over the Barrier Reef at the Cays of Albuquerque Island, Colombia

This work is a numerical model study on residual currents in vicinity of a barrier reef, induced by wind waves, the generation mechanism for the currents being radiation stress gradients in the waves. Relevant model results are extracted of typical wave directions and hights.  Results appear sound to me, but some clarifications has to be made and presentation improved.

Abstract

Fine.

  1. Introduction

Lines 56-74: Contains a very confusing description of wind and precipitation. Clarify when is the windy season, when is the dry and windy season and so on. I am sure these sections can be written more condensed. For instance, I guess the ITCZ is located furthest south in southern hemisphere summer, but don’t leave this up to the reader to guess.

  1. Materials and methods

I have not reviewed the equations.

  1. Results

Figure 6 is using geographical coordinates while figures 3-5 are using UTM coordinates. It would be easier to understand the location of the virtual buoy if they all were in UTM coordinates. Alternatively, add the virtual buoy symbol to Figure 3.

Figure 6 other issues: too small fonts, meaning of ‘Bathymetry 5km Z’? Coupling color and depth (I assume numbers are depth) is not possible to see.

Line 219: What is ‘WW-III node’?

Figure 12: Explain ‘case Hs12’ (related to Table 1?)

Line 256: ‘Figure 13 corresponds to the northeastern sector of the domain’ which domain?

Line 257: Explain meaning of ‘significant wave of 12 h during the year’. Related to Hs12?

Line 262: ‘indicating that the values with the highest steepness are on the reef’. Clarify that this is wave steepness, not bottom steepness.

Figure 13: Add the read line and letters A and B to Figure 5 as well.

Line 294: Clarify ‘Hs waves of 50%’ and ‘wave of the 50% regime and the wave of 12 h during the year’.

Line 297: I believe ‘For comparison with the results obtained in Figures 14 and 16’ should use Figure 15 in stead of 14.

Figure 17: ‘waves of the regime (solid line) and Hs12 (dashed line)’ which ‘regime’? Remove last sentence in figure legends; not appropriate to discuss results in figure legends.

  1. Conclusions

I would find it more appropriate to call this section ‘Discussion’.

Author Response

Author's Reply to the Review Report (Reviewer 2) - in the attached file

Reviewer 3 Report

the paper is good

Author Response

There were no comments

Round 2

Reviewer 1 Report

First of all, I would like to congratulate the authors for their work and dedication in the corrections. I believe the additions and modifications have greatly improved the manuscript. I have a few minor considerations but part of what I noted is "style" and in no way would I want to interfere with the style chosen by the authors.

The only detail that I think is still possible to modify and that would improve a lot, would be to include the figure that shows the position of the NOAA buoys in deep water that were used for calibration, tying this explanatory text that was very enlightening in the authors' response. One detail here - I got confused on page 10, about reference [26] which in the corrected copy doesn't appear (the reference list for me ends at 25). I did not understand if the reference Dagua Paz (Boletín Científico CIOH 2013) et al was deleted or not, but the figure is very explanatory to understand the model validation process and I think it is important.

I also suggest including the 1st explanation of the specific comments about the microtide. The explanation given here in the comments should be included in the manuscript, perhaps after line 205, just so the reader understands its effect on the parameters N1_m and N2_m as well as alpha of equation 5. In fact, the contribution of long tidal waves (lines 197 through 200) should disappear in the residual circulation (I guess).

A final point that should be discussed with the journal's production team is about the font size of the text in some figures, especially maps. Even with an excellent and "huge" monitor, I had difficulty seeing some numbers. In the final production, you may need to increase some fonts which will impact in the figure.

Thanks again for the opportunity to review this article.

Author Response

Our answers are in the attached file

Reviewer 2 Report

I am satisfied with the response the authors have given to my suggestions.

Author Response

No comments from the reviewer